# Mitochondrial surface coating with artificial lipid membrane improves the transfer efficacy

Takafumi Nakano[1,2,3], Yoshihiko Nakamura[1,3], Ji-Hyun Park[1], Masayoshi Tanaka[1] & Kazuhide Hayakawa [1✉]

Extracellular mitochondria are present and act as non-cell-autonomous signals to support energetic homeostasis. While mitochondria allograft is a promising approach in rescuing neurons, glia, and vascular cells in CNS injury and disease, there are profound limitations in cellular uptake of mitochondria together with the efficacy. Here, we modified mitochondria by coating them with cationic DOTAP mixed with DOPE via a modified inverted emulsion method to improve mitochondrial transfer and efficacy. We initially optimized the method using control microbeads and liposomes followed by using mitochondria isolated from intact cerebral cortex of male adult C57BL/6J mice. After the coating process, FACS analysis indicated that approximately 86% of mitochondria were covered by DOTAP/DOPE membrane. Moreover, the artificial membrane-coated mitochondria (AM-mito) shifted the zeta-potential toward positive surface charge, confirming successful coating of isolated mitochondria. Mitochondrial proteins (TOM40, ATP5a, ACADM, HSP60, COX IV) and membrane potentials were well maintained in AM-mito. Importantly, the coating improved mitochondrial internalization and neuroprotection in cultured neurons. Furthermore, intravenous infusion of AM-mito immediately after focal cerebral ischemia-reperfusion amplified cerebroprotection in vivo. Collectively, these findings indicate that mitochondrial surface coating with artificial lipid membrane is feasible and may improve the therapeutic efficacy of mitochondria allograft.

[1] Neuroprotection Research Laboratory, Departments of Radiology and Neurology, Massachusetts General Hospital and Harvard Medical School, Charlestown, MA 02129, USA. [2] Department of Physiology and Pharmacology, Faculty of Pharmaceutical Sciences, Fukuoka University, Fukuoka 814-0180, Japan. [3] These authors contributed equally: Takafumi Nakano, Yoshihiko Nakamura. ✉email: khayakawa1@mgh.harvard.edu

Mitochondria are energetic components of cells and essential for maintaining cellular function in mammals[1] through regulating levels of adenosine triphosphate (ATP)[2], fatty acids[3], and cellular calcium[4]. In a context of central nervous system (CNS) disease, accumulating mitochondrial ROS and inflammasome along with imbalanced mitochondrial membrane permeability may cause progression of cell death and neuroinflammation[5]. Therefore, restoring mitochondrial perturbation within cells is a major therapeutic strategy in many CNS disorders including stroke[6], hemorrhagic stroke[7], spinal cord injury[8], Alzheimer's disease (AD)[9], and Parkinson's disease (PD)[10].

It has been reported that mitochondria are present in extracellular space and participate in non-cell autonomous mechanisms in the CNS[11,12]. Given that endogenous mitochondrial transfer may support metabolic homeostasis in damaged neurons[13], exogenous mitochondrial transplantation may become a new therapeutic intervention to promote neuroprotection and recovery in various CNS disorders[8,14–16]. However, it is important to note that naked mitochondria or mitochondrial surface lipid can be deleterious factors. For example, oxidized cardiolipin on mitochondrial surface may activate pro-inflammatory programs in leukocytes[17]. In this proof-of-concept study, we attempted to modify mitochondrial surface with cationic and fusogenic lipids, DOTAP and DOPE[18,19], and aimed to minimize the direct contact of mitochondrial components to cells while maximizing effects of mitochondrial transfer.

## Results

**Optimization of the inverted emulsion method to decorate particle surface with DOTAP and DOPE.** Our strategy of coating mitochondria—MitoCoat with artificial cationic and fusogenic lipids (AM-mito) was meant to improve mitochondria delivery and cerebroprotection efficacy (Fig. 1a). The inverted emulsion method has been well established to produce giant unilamellar vesicles in physiological buffer and encapsulate various molecules from genetic materials to biological macromolecules[20]. First, we optimized total volumes of water-in-oil emulsion, buffer, concentrations, and ratio of DOTAP and DOPE, temperature, and speed while extracting encapsulated materials into PBS during centrifuge (Fig. 1b). We initially attempted to validate encapsulation using 1% evans blue solution. When the production of liposomes was unsuccessful, the evans blue was visually leaked to PBS solution along with lesser amount of pellets precipitated in the bottom of tube. After various conditions were tested, evans blue (5 µL of 1%)-incorporated liposomes were successfully produced in 1 mM of DOTAP/DOPE (1:1) after a centrifuge at 4000 $g$ for 10 min at 4 °C (Fig. 1c).

Next, we utilized microbeads (~1 µm) or liposomes (~100 nm) to test the ability of the inverted emulsion method to cover various sizes of materials with cationic lipid membrane (Fig. 1d). In a use of microbeads, evans blue dye appeared to be incorporated between microbeads and DOTAP/DOPE membrane after coating process while microbeads mixed with evans blue without the process did not show the fluorescent signals (Fig. 1e). Furthermore, we also attempted to encapsulate liposomes in the DOTAP/DOPE vesicles. Excitingly, size of the liposomes shifted from 100 nm to approximately 205.1 nm (Fig. 1f) and surface charge of the particles changed from −50 mV to 45 mV after extracting vesicles (Fig. 1g), indicating the encapsulation of liposomes within cationic membrane. Collectively, the optimized inverted emulsion method can encapsulate large and small particles ranging from 100 nm to 1 µm.

**Mitochondrial surface coating with DOTAP and DOPE.** Then, we used mitochondria as the third material. Mitochondria were

isolated from intact cerebral cortex of male C57BL/6J mice and we prepared mitochondrial suspension in mitochondria functioning buffer for the inverted emulsion-mediated vesicle encapsulation. Notably, FACS analysis revealed that approximately 86% of mitochondria were coated by DOTAP/DOPE together with evans blue while other 14% of mitochondria barely showed evans blue signal (Fig. 2a). This may be because of no incorporation of evans blue within particles or unexpected failure to encapsulate mitochondria. Moreover, the averaged particle size was slightly increased (Fig. 2b) along with the positive shift of mitochondrial surface charge (Fig. 2c). Importantly, western blot and FACS analysis demonstrated that the mitochondrial coating with DOTAP/DOPE did not affect mitochondrial proteins including TOM40, ATP5A, ACADM, HSP60 and COX IV (Fig. 2d). Moreover, the process of surface modification did not significantly affect ATP content (nmol per mg mitochondria protein) and mitochondrial shape (Fig. 2e–f), while FACS analysis revealed that the coating process improved purity of mitochondria (Fig. 2g) and stabilized mitochondrial membrane potentials (Fig. 2h).

**Mitochondrial coating with DOTAP/DOPE improved neuroprotective efficacy after transfer.** So far, our data implicate that coating mitochondria with DOTAP/DOPE artificial membrane may be feasible. But, an important question is whether AM-mito transfer improves neuroprotective efficacy. To address the issue, mouse primary neurons were subjected to oxygen-glucose deprivation (OGD) for 2 h, then control mitochondria or AM-mito were added onto the injured neurons and evaluated mitochondrial transfer and neuroprotection (Fig. 3a). At 3 h post-transfer of AM-mito, levels of intracellular mitochondria and mitochondrial protein ATP5A appeared to be higher in the recipient neurons in comparison of neurons treated with control mitochondria (Fig. 3b,c). Furthermore, AM-mito significantly improved neuroprotection analyzed by water-soluble tetrazolium salt (WST)-based assay against oxygen-glucose deprivation in vitro (Fig. 3d). As a negative control, DOTAP/DOPE vesicles were not significantly protective when they were treated in neurons subjected to 2 h OGD (Fig. 3e), suggesting that the AM-mito entry into neurons may generate benefits beyond cell-cell communication through the artificial cationic membrane. Moreover, Mitotracker Green-labeled AM-mito (10 µg per well) were found in approximately 55% of total neurons at 24 h after OGD and 46% or 8.7% of AM-mito+ neurons were survived or dying/dead, respectively (Fig. 3f). This may further support that AM-mito incorporation into neurons may accommodate protection against ischemic injury.

To further validate the beneficial neuroprotection by AM-coated mitochondria transfer, C57Bl6 male mice were subjected to 60 min of transient focal cerebral ischemia. Mouse brain-derived mitochondrial fractions (100 µg protein per 100 µL PBS per mouse) were coated with DOTAP/DOPE vesicles. We also prepared uncoated control mitochondria that were processed through the inverted emulsion method without DOTAP/DOPE lipids. Then, control mitochondria or AM-mito were infused intravenously immediately after reperfusion with full blinding and randomization (Fig. 3g). Fluorescent imaging confirmed that transplanted AM-mito were more found in the ipsilateral hemisphere compared to control mitochondria at 2 h after treatment. (Fig. 3h). After 72 h, mice were sacrificed and brains removed for analysis. Excitingly, treatment with AM-mito showed better neuroprotective effect compared to control mitochondria (Fig. 3i).

## Discussion

In this study, we found that (i) the inverted emulsion method could be utilized for large particles including mitochondria for

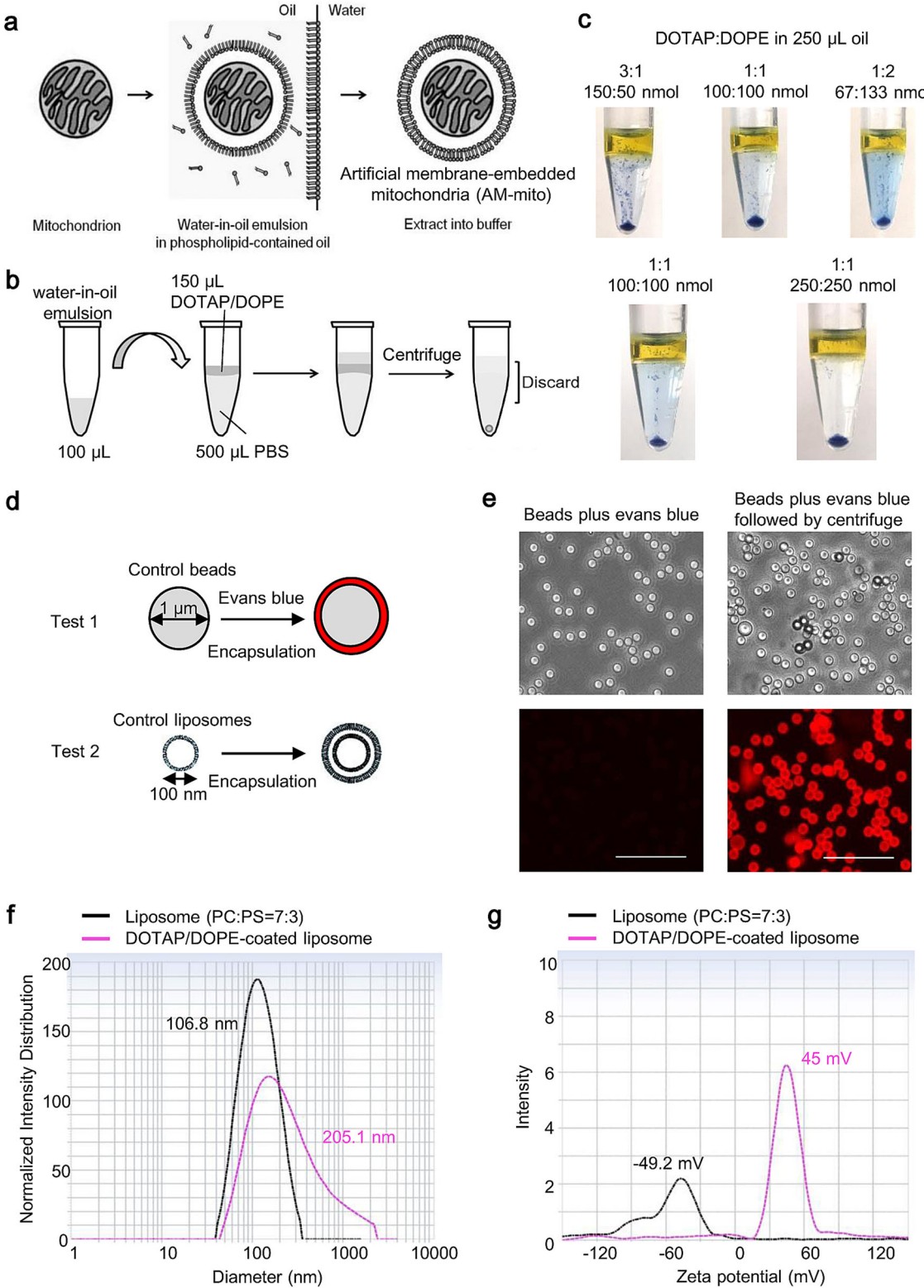

their surface modification, (ii) artificial cationic membrane-coated mitochondria (AM-mito) maintained their proteins and membrane potentials, and (iii) treatment with AM-mito improved mitochondrial transfer and neuroprotection efficacy.

Mitochondrial transfer has been proposed as a new paradigm for interdependent cell-cell signaling in the CNS[21,22]. It is known that extracellular mitochondria and its components act as damage-associated molecular patterns (DAMPs) and induce inflammation[23]. Moreover, intercellular mitochondrial transfer regulates neuroprotection or neurodegeneration, depending on the context[13,24]. What remains missing from the collective literature to date is how mitochondria become beneficial signals and promote neuroprotection. Our present study provides proof-of-principle that mitochondrial encapsulation using cationic lipid DOTAP mixed with DOPE may improve neuroprotection mediated by mitochondrial transfer.

**Fig. 1 Optimization of MitoCoat process using the inverted emulsion method. a** Overview of the inverted emulsion method. **b** The experimental design for biomaterial encapsulation. Materials dissolved in water-based buffer are mixed with 100 μL of mineral oil containing DOTAP/DOPE followed by generating water/ oil (W/O) emulsion by gently pipetting 10 times. The W/O emulsion is transferred onto 150 μL of mineral oil or olive oil (only for initial assessments) containing DOTAP/DOPE on 500 μL of PBS prepared in 1.5 mL tube at least 30 min before adding W/O emulsion. Five minutes after slowly adding W/O emulsion, the test tubes are spinned by the centrifugation. Supernatant was discarded and the pellet is carefully resuspended in PBS and washed one time. **c** After various conditions were tested, evans blue (5 μL of 1%)-incorporated liposomes were successfully produced in 1 mM of DOTAP/DOPE (1:1) after a centrifuge at 4000 g for 10 min at 4 °C. **d** Scheme of the inverted emulsion encapsulation for microbeads (~1 μm) and liposomes (~100 nm). **e** Microbeads (Test1) were encapsulated by DOTAP/ DOPE vesicles along with evans blue. Evans blue dye was successfully incorporated between microbeads and DOTAP/DOPE membrane after encapsulation while microbeads mixed with evans blue without the process did not show the fluorescent signals. Scale: 10 μm. **f** We attempted to encapsulate liposomes compose of 7:3 molar ratio of L-alpha-phosphatidylcholine: L-±-phosphatidylserine (Test2) in the DOTAP/DOPE vesicles. Nano Delsa showed that size of the liposomes shifted from 100 nm to approximately 205.1 nm. **g** Zeta potential of particles changed from −50 mV to 45 mV following DOTAP/DOPE encapsulation.

While more than 400 clinical trials for mitochondrial-targeted medical intervention including completed trials have been registered at ClinicalTrials.gov, medicines targeting mitochondrial energy production remain to be developed. As a new mitochondria-targeted therapy, exogenous mitochondrial transplantation has been emerged and tested in models of various CNS injuries or diseases[25]. But, it is important to note that there are profound limitations such as mitochondrial delivery to target cells and preventing toxicity of mitochondrial components, thus modifying mitochondria may be essential to take into account for clinical translation. It has been reported the feasibility of mitochondrial modification. For instance, Pep-1 conjugation to mitochondria improved mitochondrial delivery into dopaminergic neurons and showed better behavioral outcomes compared to ones treated with unmodified control mitochondria in a mouse model of Parkinson's disease[26]. Moreover, transplantation of polymer-functionalized isolated mitochondria into cardiac cells improved cellular internalization and oxygen consumption rate[27]. Cationic liposomes are traditionally utilized for the delivery of genetic materials because cationic lipids are capable of neutralizing the negative charge of plasmid DNA with positive charge and increase efficiency to capture plasmid and deliver them into cells. This concept can be applied to the delivery of energetic mitochondria that are negatively charged[28]. Moreover, covering mitochondrial surface may also prevent to expose mitochondrial components that may induce unwanted detrimental effects to cells. Therefore, our idea of encapsulation of isolated mitochondria may be one to improve mitochondrial delivery along with reducing toxicity.

Nevertheless, there are a few caveats that warrant further investigation. First, we used brain mitochondria for encapsulation. But, respiratory active mitochondria are able to isolate from various clinically relevant sources including skeletal muscle, placenta, iPSC, and platelets. Whether the inverted emulsion method can be applied to other mitochondria should be addressed to seek clinical translation. Second, whether mitochondria coating with DOTAP/DOPE also preserves their functional property remains to be fully investigated. In this study, DOTAP/DOPE-coated mitochondria showed higher membrane potential, better incorporation into cells in accompanied by more remaining in the ipsilateral hemisphere compared to non-coated control mitochondria, implicating that MitoCoat may stabilize isolated mitochondrial function. Regarding clinical translation, mitochondria preservation for long-term storage significantly increases flexibility of the transplantation therapy yet it is the biggest challenge. It has been reported that glycerol, DMSO, or trehalose may preserve mitochondrial integrity following freeze and thaw process. Therefore, it is worth investigating whether MitoCoat supports long-term storage along with traditional methods utilizing glycerol (20%)[29], DMSO (20–50%)[29], or trehalose (250–300 mM)[30]. Third, higher concentration of isolated mitochondria showed lower ability of neuroprotection. Therefore, it is

critical to verify the optimal concentration of mitochondria that does not induce toxicity yet produces beneficial protection when it comes to the treatment with mitochondria-enriched fraction. On contrary, DOTAP/DOPE-encapsulation supported neuroprotection in a higher dose of mitochondria-enriched fraction. Moreover, AM-coated mitochondria amplified cerebroprotection in a mouse model of focal ischemia. Although artificial membrane coating may be promising, neuroprotection mediated by AM-mito was peaked in the treatment of 50 μg per well ($1 × 10^5$ cells) in vitro, suggesting that the proper amount of mitochondria per cell should be addressed to maximize beneficial cerebroprotection in future studies. Fourth, while mitochondrial encapsulation maintained mitochondrial proteins, ATP, and membrane potentials, a long term activity of mitochondria and the resistance to pathological environments such as high calcium, ionic imbalance, and high ROS remain to be examined. Fifth, although mitochondrial allograft may explicit beneficial protection under pathophysiological conditions, targeting specific tissues or cells are still challenging. The mitochondrial matrix is highly negatively charged[31], suggesting that transplanted mitochondria may be attracted to acidotic ischemic tissue[16]. But what remains to be elucidated is how to target dying cells that should be selectively rescued. It has been considered that the exposure of phosphatidylserine (PS) on the outer plasma membrane are a hallmark of dying cells. Because PS is negatively charged, mitochondria coated by cationic artificial membrane might support electrostatic interaction. Further studies are warranted to rigorously validate mitochondria migration and incorporation into brain cells such as neurons, astrocytes, microglia, and endothelial cells under pathophysiological conditions in the CNS. Finally, underlying mechanisms of AM-mito incorporation into neurons remains to be fully defined. Among the many postulated mechanisms of neuronal internalization of mitochondria include extracellular mitochondria can enter cells via endocytosis[32], integrin-dependent mechanisms[13] or macro-pinocytosis[28], whether DOTAP/DOPE-encapsulation affects these entry mechanisms should be addressed in future studies.

Mitochondrial function is critical for CNS recovery after injury or disease[33]. Within the emerging paradigm of mitochondrial transplantation, our study suggests that mitochondrial encapsulation within DOTAP/DOPE vesicles may improve mitochondrial transfer-mediated cerebroprotection. Further studies are warranted to dissect and validate the in vivo consequences of this phenomenon in animal models of brain injury and neurodegeneration.

## Methods

**Animal study.** All experiments were performed following an institutionally approved protocol in accordance with National Institutes of Health guidelines and with the United States Public Health Service's Policy on Human Care and Use of Laboratory Animals and following Animals in Research: Reporting In vivo Experiments (ARRIVE) guidelines.

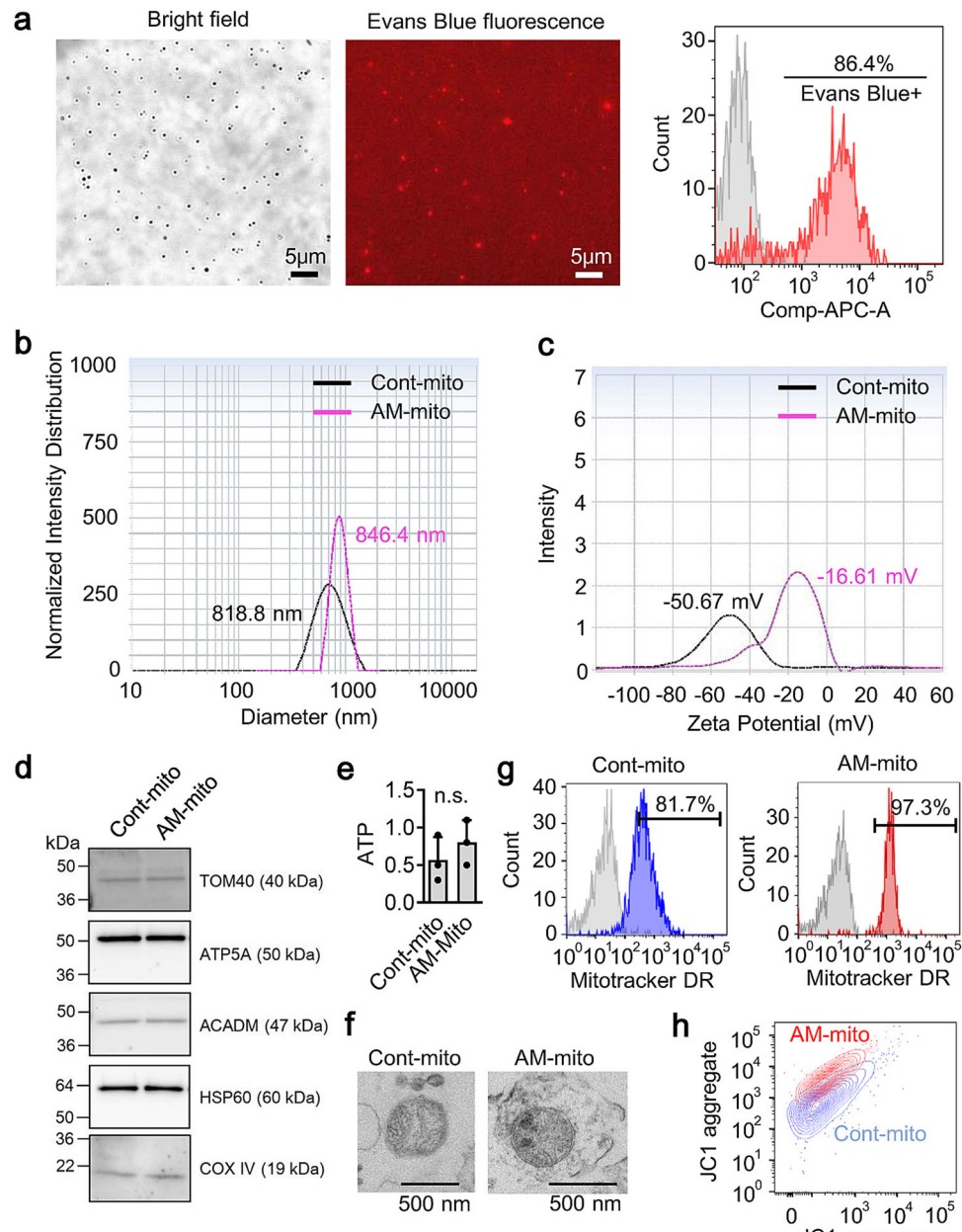

**Fig. 2 MitoCoat with cationic lipid membrane. a** Mitochondria were isolated from intact cerebral cortex of male C57BL/6 J mice and suspended in mitochondria functioning buffer. Then the inverted emulsion method was applied to them. FACS analysis revealed that approximately 86% of mitochondria were wrapped by DOTAP/DOPE vesicles. Evans blue was mixed with mitochondrial functioning buffer to track mitochondrial encapsulation. **b** Particle size of artificial membrane-coated mitochondria (AM-mito) was approximately 846.4 nm. **c** After coating, zeta potential of mitochondria was changed from −50.67 mV to −16.61 mM. **d** Western blot confirmed that mitochondrial proteins such as TOM40, ATP5A, ACADM, HSP60, and COX IV were well preserved in AM-mito equivalent to control mitochondria. **e** ATP content (nmol per mg protein) was not different between control mitochondria and AM-mito ($n = 3$). $P = 0.3987$, unpaired $t$-test (two-tailed). Results were expressed as mean ± SD. **f** TEM images showed no clear morphological change in mitochondria following artificial membrane coating. **g** FACS analysis showed higher purity of mitochondria following coating process around 97% in comparison of control mitochondria around 81%. **h** FACS analysis showed that JC1 mitochondrial membrane potential became slightly higher in AM-mito compared to control mitochondria.

**Primary neuron cultures**. Primary neuron cultures were prepared from cerebral cortices of E18-day-old Sprague-Dawley rat or C57BL/6J mouse embryos. Briefly, cortices were dissected and dissociated using papain dissociation system (Worthington Biochemical Corporation, LK003150). Cells were spread on plates coated with poly-D-lysine (Sigma, P7886) and cultured in Dulbecco's modified Eagle medium (NBM, Life Technology, 11965-084) containing 25 mM glucose, 4 mM glutamine, 1 mM sodium pyruvate, and 5% fetal bovine serum at a density of $2 \times 10^5$ cells/mL (1 mL for 12 well format, 0.5 mL for 24 well format). At 24 h after seeding, the medium was changed to Neurobasal medium (Invitrogen, 21103-049) supplemented with B-27 (Invitrogen, 17504044) and 0.5 mM glutamine. Cells were cultured at 37 °C in a humidified chamber of 95% air and 5% $CO_2$. Over 95% of purity of neuron cultures was determined by MAP2 staining in our previous study.

**Oxygen-glucose deprivation (OGD) and reoxygenation**. OGD experiments were performed using a specialized, humidified chamber (Heidolph, incubator 1000, Brinkmann Instruments, Westbury, NY) kept at 37 °C, which contained an anaerobic gas mixture (90% $N_2$, 5% $H_2$, and 5% $CO_2$). To initiate OGD, culture medium was replaced with deoxygenated, glucose-free Dulbecco's modified Eagle medium (Life Technology, 11966-025). After 2 h challenge, cultures were removed from the anaerobic chamber, and the OGD solution in the cultures was replaced with maintenance medium. Cells were then allowed to recover for 24 h for neurotoxicity assay.

**In vivo focal ischemia model**. Male C57BL6 mice (11–12 weeks, Jackson Laboratories) were deeply anaesthetized with isoflurane (5% to 1.5%) in 30%/70%

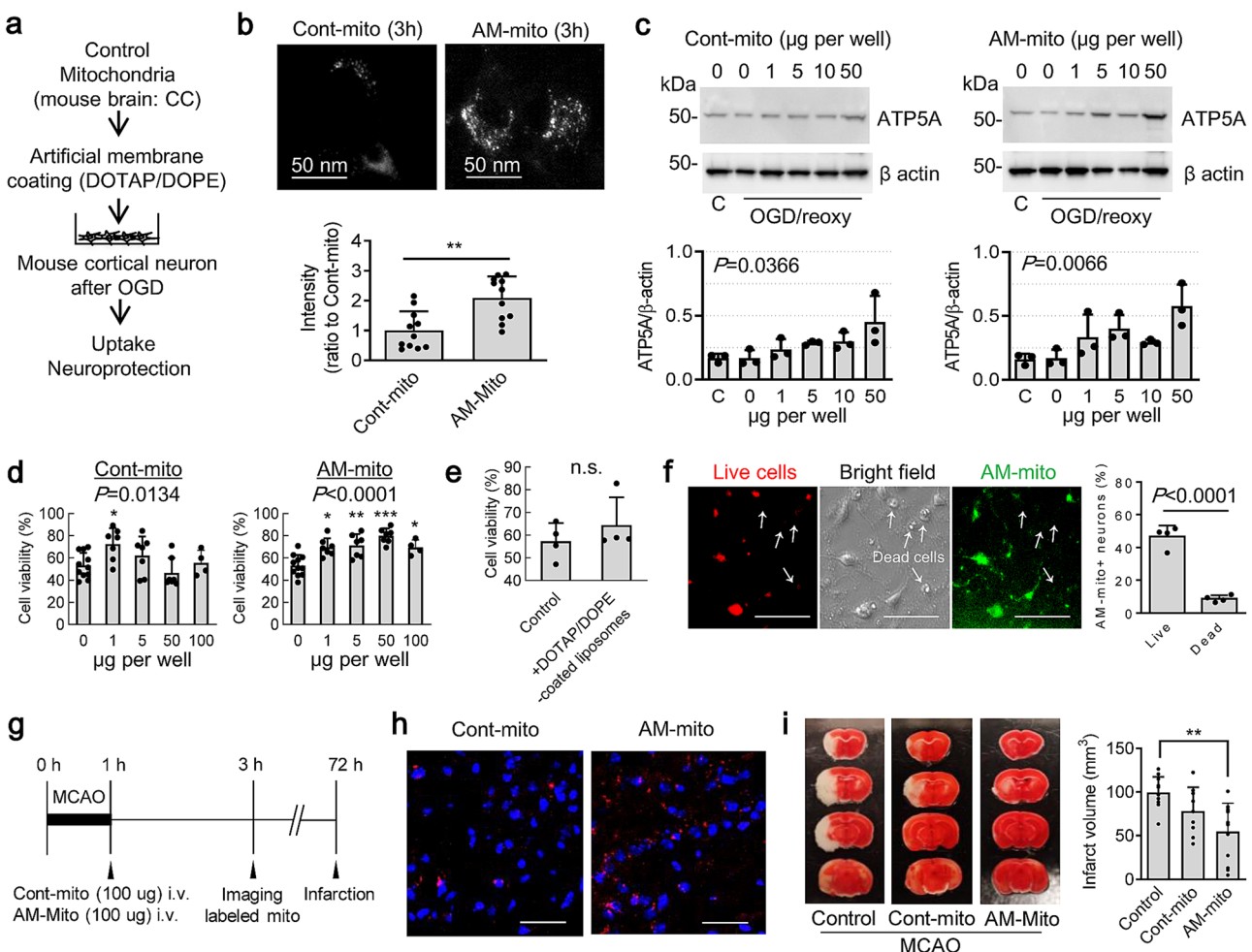

**Fig. 3 DOTAP/DOPE-coated mitochondria improved the internalization and neuroprotection. a** Mouse primary neurons were subjected to oxygen-glucose deprivation (OGD) for 2 h, then control mitochondria or AM-mito were added onto the injured neurons and evaluated mitochondrial transfer and neuroprotection. **b** Mitotracker Deep Red (200 nM) was used to label mitochondria before encapsulation. At 3 h post-transfer, AM-mito (5 µg) were incorporated into the recipient neurons (Cont-mito: $n = 11$, AM-mito: $n = 11$). **$P = 0.0013$, unpaired t-test (two-tailed). **c** Western blot confirmed that AM-mito-treated neurons showed higher mitochondrial protein (ATP5A) compared to ones treated with control mitochondria. Cont-mito ($n = 3$): $P = 0.0366$, AM-mito ($n = 3$): $P = 0.0066$, One-way ANOVA. **d** Neuronal viability assay was performed by water-soluble tetrazolium salt (WST)-based assay at 24 h post-OGD. Treatment with AM-mito improved cell survival in a concentration dependent manner (Cont-mito: 0 µg: $n = 12$, 1 µg: $n = 7$, 5 µg: $n = 7$, 50 µg: $n = 7$, 100 µg: $n = 4$, AM-mito: 0 µg: $n = 12$, 1 µg: $n = 7$, 5 µg: $n = 7$, 50 µg: $n = 7$, 100 µg: $n = 4$). *$P < 0.05$, **$P < 0.01$, ***$P < 0.001$ vs 0. One-way ANOVA followed by Tukey's test. **e** DOTAP/DOPE-coated liposomes did not protect neurons subjected to OGD (OGD: $n = 4$, DOTAP/DOPE-coated liposomes: $n = 4$). $P = 0.3660$, unpaired t-test (two-tailed). **f** Rat primary neurons were subjected to oxygen-glucose deprivation (OGD) for 2 h, then Mitotracker Green-labeled AM-mito were added onto the injured neurons. Mitotracker Green-labeled AM-mito (10 µg per well) were found in approximately 55% of total rat primary neurons at 24 h after OGD and 46% or 8.7% of AM-mito-positive neurons were survived or dying/dead neurons, respectively ($n = 4$). $P < 0.0001$, unpaired t-test (two-tailed). Scale: 100 µm. **g** Male C57BL6 mice were subjected to 60 min transient focal cerebral ischemia. Treatment with control mitochondria (100 µg/100 µL) or AM-mito (100 µg/100 µL) was performed right after reperfusion with fully blinding. Brain infarction was determined at 72 h after focal cerebral ischemia. **h** Mitochondria were labeled by Mitotracker Deep Red (DR) (100 nM) before intravenous infusion. Infused AM-coated mitochondria were more found in ipsilateral cortex compared to control mitochondria at 2 h after transient focal cerebral ischemia - reperfusion. Scale: 100 µm. **i** AM-mito significantly decreased brain infarction compared to untreated group (no treatment control: $n = 11$, Control mitochondria: $n = 9$, AM-mito: $n = 12$). Control vs AM-mito: **$P = 0.0011$, One-way ANOVA ($P = 0.0017$) followed by Tukey's test. All results were expressed as mean ± SD.

oxygen/nitrous oxide. After midline skin incision, 8-0 nylon monofilament coated with silicon resin was introduced through a small incision into the common carotid artery. Adequate cerebral ischemia was assessed by Laser Doppler flowmetry and by examining forelimb flexion after the mice recovered from anesthesia. Sixty minutes after occlusion, the mice were re-anesthetized, and reperfusion was established by withdrawal of the filament.

**Cell viability assays.** Cell proliferation was assessed by Water-soluble tetrazolium salt (WST) reduction assay (Dojindo), which detects dehydrogenase activity of viable cells. The cells were incubated with 10% WST solution for 1 h at 37 °C. Then the absorbance of the culture medium was measured with a microplate reader at a test wavelength of 450 nm and a reference wavelength of 630 nm.

**ATP measurement.** mitochondrial ATP was determined by CellTiter-Glo luminescence (Promega). Standard ATP was used for measurement of ATP content (nmol/mg protein) in control mitochondria or AM-mitochondria.

**Electron microscopy analysis.** Following a centrifugation at 14,000 rpm for 10 min, pellets were fixed in 2.0% glutaraldehyde in 0.1 M sodium cacodylate buffer, pH 7.4 (Electron Microscopy Sciences) for one hour at room temperature on a rocker. After the process of embedding the pellet, thin sections were cut on a Leica EM UC7 ultramicrotome, collected on formvar-coated grids, stained with uranyl acetate and lead citrate, and examined in a JEOL JEM 1011 transmission electron microscope at 80 kV. Images were collected using an AMT digital imaging system (Advanced Microscopy Techniques).

**Mitochondrial isolation**. Mitochondrial enriched fractions were obtained from intact cerebral cortex in male C57BL/6 J mice or male Sprague-Dawley rats and suspended in mitochondria isolation buffer—10 mM HEPES (pH7.5) buffer containing 25 mM sucrose, 1 mM ATP, 0.1 mM ADP, 5 mM sodium succinate, and 2 mM $K_2HPO_4$. Briefly, each tissue was homogenized in glass tissue grinders in mitochondria isolation buffer at 4 ºC. Following removal of cellular debris with repeated centrifugation at 1000 $g$ at 4 ºC for 5 min followed by 4000 $g$ at 4 ºC for 5 min. Then, obtained supernatant was further spinned down by the centrifugation at 8000 $g$ at 4 ºC for 10 min to obtain mitochondria-enriched pellet.

**Mitochondria membrane potential measurement**. To monitor mitochondrial health, JC-1 dye (invitrogen, T-3168) was used to assess mitochondrial membrane potential. Isolated mitochondria were incubated with JC1 (0.8 μM) for 30 min at 37 °C. Mitochondria membrane potential was determined by FACS by BD Fortessa. FACS analysis was performed using an unstained or phenotype control for determining appropriate gates, voltages, and compensations required in multivariate flow cytometry.

**Western blot analysis**. Each sample was loaded onto 4–20% Tris-glycine gels. After electrophoresis and transferring to nitrocellulose membranes, the membranes were blocked in Tris-buffered saline containing 0.1% Tween 20 and 0.2% I-block (Tropix, T2015) for 90 min at room temperature. Membranes were then incubated overnight at 4 °C with following primary antibodies, anti-β-actin (1:1000, Sigma-aldrich A5441), anti-TOM40 (1:200, Santacruz, sc-11414), anti-ATP5A (1:500, Abcam, ab14748), anti-ACADM (1:500, Abcam, ab92461), and anti-HSP60 (1:500, Abcam, ab46798). After incubation with peroxidase-conjugated secondary antibodies, visualization was enhanced by chemiluminescence (GE Healthcare, NA931-anti-mouse, or NA934- anti-rabbit, or NA935- anti-rat). Optical density was assessed using the NIH Image analysis software.

**Coating with DOTAP/DOPE**. To coat microbeads (F13838, 1 μm, Thermo-FisherScientific), liposomes compose of 7:3 molar ratio of L-alpha-phosphatidylcholine: L-±-phosphatidylserine (Encapsula NanoScience), or isolated mitochondria, the inverted emulsion method was used. Mitochondria were isolated from intact cerebral cortex of male C57BL/6J mice (12-14 weeks) or male Sprague-Dawley rats (12–14 weeks) and we prepared mitochondrial suspension in mitochondria buffer consist of 10 mM HEPES pH 7.5, 250 mM Sucrose, 1 mM ATP, 0.1 mM ADP, 5 mM Sodium succinate, 2 mM Dipotassium phosphate, and 1% polyvinyl alcohol. Mitochondrial suspension (up to 50 μg/5 μL) was mixed with 100 μL of mineral oil containing 1 mM DOTAP/DOPE (1:1), followed by generating water/oil (W/O) emulsion by gently pipetting 10–15 times. For initial evaluation to develop the inverted emulsion method, we used olive oil and evans blue to visualize the layers of oil and encapsulation of evans blue within liposomes. The W/O emulsion was transferred onto 150 μL of mineral oil containing 1 mM DOTAP/DOPE (1:1) on 500 μL of PBS prepared in 1.5 mL tube at least 30 min before adding W/O emulsion. Five minutes after slowly adding W/O emulsion, the test tubes were spinned by the centrifugation at 4000 $g$ for 10 min at 4 °C. Supernatant was discarded and the pellet was carefully resuspended in mitochondria buffer and washed one time with a centrifugation with 4000 g for 5 min at 4 ºC. Mitochondrial quality after coating was assessed by particle size and zeta potential analysis by Delsa Nano, western blot and FACS.

**Statistics and reproducibility**. Results were expressed as mean ± SD. All of experiments were performed with full blinding, allocation concealment and randomization. When only two groups were compared, unpaired t-test (two-tailed) was used. Multiple comparisons were evaluated by one-way ANOVA followed by Tukey's test. $P < 0.05$ was considered to be statistically significant.

**Reporting summary**. Further information on research design is available in the Nature Research Reporting Summary linked to this article.

## Data availability

Uncropped western blot images are presented in Supplementary Figure 1. The source data for all the graphs presented in figures are available in Supplementary Data 1.

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

## Acknowledgements
This research was funded by NIH NINDS grant numbers NS094756, NS126280, and MGH ECOR Interim Support Fund. The authors thank Sandra Nakasone for assistance with the Nano Delsa analysis, performed in the Center for Nanoscale Systems at Harvard University. The authors also thank Eng H. Lo for many helpful discussions.

## Author contributions
Conceptualization, K.H.; conducted experiments/data analysis, T.N., J.H.P., MT., Y.N., K.H.; Writing-Original Draft Preparation, T.N., J.H.P., K.H.; Funding Acquisition, K.H.

## Competing interests
The authors declare no competing interests.
