## [Peer Review File · Communications Biology]

Reviewers' comments:

Reviewer #1 (Remarks to the Author):

This study builds upon the same authors' pioneering report (Hayakawa et al., Nature 2016) that mitochondria transfer occurs after stroke representing a host neuroprotective strategy. Here, the authors advance the thesis that optimizing mitochondrial transfer may allow a much better neuroprotective outcome. The major claims of the paper involve a novel technology of coating the mitochondria with cationic DOTAP mixed with DOPE through an emulsion method resulting in enhanced mitochondrial transfer and efficacy. The technique successfully leads to about 86% artificial membrane-coated mitochondria (AM-mito) with a positive surface charge. These AM-mito aided cellular internalization and significantly protected neurons from the in vitro stroke insult of oxygen glucose deprivation (OGD). These findings provide an innovative tool in the application of mitochondria-based regenerative medicine for stroke and other neurological disorders. This approach further strengthens the central role of mitochondria as the powerhouse of the cell and targeting these key organelles add novel concepts in our understanding of disease pathology and treatment. The study contains straightforward statistical analysis, detailed methodology to replicate the work, and logical data interpretations and conclusions. My comments below are mostly technical that will require only clarifications from the authors.

1. Abstract: Best to identify the source of the mitochondria - were they derived from neurons?

2. Abstract: Briefly acknowledge that while stroke impairs the neurovascular unit, comprising of neurons, astrocytes, and other cell types, the present study focusing on neurons subjected to OGD represents a proof-in-principle to probe mitochondrial transfer which is likely to occur not only in neurons, but also in astrocytes, endothelial cells, etc.

3. Introduction: In referencing repair of the mitochondrial functions for stroke, Rutkai and colleagues specifically demonstrate impaired mitochondrial oxygen consumption rate after stroke. A better reference for restoring mitochondrial function in stroke may be that of Nguyen et al (Eye Opener in Stroke. Stroke. 2019;50(8):2197-2206. doi:10.1161/STROKEAHA.119.025249).

4. Methods and Results: Briefly stating the possible reason for why the 14% neurons escaped AM-mito coating is welcomed. Also, it will be interesting if there's a way to delineate whether OGD preferentially damaged neurons which were not AM-mito coated as opposed to those that were AM-mito may be interesting as this will further support the claim that the AM-mito technology confers neuroprotection.

5. Discussion: The statement "It has reported that extracellular mitochondria can enter cells via endocytosis 29, integrin-dependent mechanisms 13 or macro-pinocytosis 28" may need to be re-written as follows: ". Among the many postulated mechanisms of neuronal internalization of mitochondria include extracellular mitochondria can enter cells via endocytosis 29, integrin-dependent mechanisms 13 or macro-pinocytosis 28."

v

Reviewer #2 (Remarks to the Author):

The manuscript by Nakamura and colleagues described a method to modify isolated mitochondrial surface to improve its cellular transfer efficacy. It has been proposed that mitochondrial allograft is a new approach for the intervention of cell damage and neurodegeneration, however, the efficient delivery of mitochondria into the cells is a big technique obstacle. The authors set to address this issue by trying to use biomaterial encapsulation to help mitochondria to be transferred into the cell, and use

a OGD assay to assess the efficiency and consequence in a prove of concept manner. In general, the study is technique-driven, novel and important. However, regarding to the data quality, I have the following concerns:

1. Fig. 1c, what is the criteria to judge the optimal ratio for DOTAP:DOPE? The authors did not explain how to read the data, is any kind of quantitation needed here?
2. Fig. 2d, to prove the mitochondria itself was not changed after surface modification, the study used some mitochondria protein as the readout. Is there any other functional readout, like ATP synthesise, morphology changes to be detected?
3. Fig. 3b, it would be nice to present a quantitation here to better show the increased intracellular mitochondria. Moreover, a negative control that treatment of only coating material ()DOTAP/DOPE) is needed here to show the effect of those biomaterials on intracellular endogenous mitochondria.
4. Fig. 3d, what is WST assay stand for? There is not any full name presented throughout the manuscript.
5. Since the ultimate goal of this study is for the intervention of the disease, it would be nice to talk about their thoughts about in vivo delivery of mitochondria into specific region and cell types in the discussion section.
6. Minor point: Main text line 4, is there a typo "and progression of cell death and 5"

Reviewer #3 (Remarks to the Author):

This study explore the possibility of applying artificial lipid membranes on the surface of mitochondria to improve effectiveness of mitochondrial transplantaion. The idea is novel and interesting, and has clinical significance for the regulation of neuroprotection or neurodegeneration. Overall, the manuscript confirms that the optimized inverted emulsion method is feasible. The mitochondria were then isolated from the cerebral cortex of C57BL/6J mice and constructed into AM-mito. It was verified that the mitochondria could be encapsulated successfully, the mitochondrial protein and membrane potential were maintained. In neurons exposed to OGD, AM-mito improved mitochondrial transfer and cytoprotection efficacy. The experimental design of this study is reasonable, convincing, logical, and highlights potential problems that deserve consideration. But, some issues need to be addressed as below:

1. Fig.2d should include mitochondria internal reference protein such as COX IV in order to confirm that the total amount of mitochondria is the same in both groups.
2. It is better that Fig.3c add statistic diagram to compare the transfer efficiency of Cont-mito and AM-mito.
3. Please add the specific methods of extracting and isolating mitochondria in the part of Materials and Methods.
4. Mitochondrial transplantaion as a potential neuroprotective therapy, the safety problems associated with their mitochondrial preservation and transport in vitro cannot be ignored. Can the surface coating of artificial lipid membrane improve the stability of mitochondria in vitro? An explanation or argument can be added in the "discussion" section.
5. Higher concentration of isolated mitochondria may slightly increase neuron death. While improving the transport efficiency, may AM-mito also have potential neurotoxicity? In Fig.3d, the group with a higher concentration of AM-mito can be added to discuss this issue.
6. In addition to the caveats listed in the discussion, the application mode of coated mitochondria should be explored. Different from endogenous mitochondrial transfer, it is difficult to fulfill the transplantaion of exogenous liposome coated mitochondria to targeted tissue or cells. It is recommended to add this part to the discussion.
7. The resolution of figures (the text part mainly) is low, especially in Fig.2a and Fig.2e.

Resubmission of COMMSBIO-22-1227-T, Nakano and Nakamura et al.

Reviewer #1:

1. Abstract: Best to identify the source of the mitochondria - were they derived from neurons?

Response: We apologize for being unclear on the source of mitochondria, and thank the reviewer for giving us another opportunity to clarify it. Mitochondria were isolated from intact cerebral cortex of male C57BL/6J mice (12-14 weeks) throughout this study. We now included the information in our main document and method.

2. Abstract: Briefly acknowledge that while stroke impairs the neurovascular unit, comprising of neurons, astrocytes, and other cell types, the present study focusing on neurons subjected to OGD represents a proof-in-principle to probe mitochondrial transfer which is likely to occur not only in neurons, but also in astrocytes, endothelial cells, etc.

Response: Thank you for the suggestion. We tried to include this rationale of neurovascular cells and the potential of AM-mito protecting a wide range of cells in the revised Abstract and discussion sections.

3. Introduction: In referencing repair of the mitochondrial functions for stroke, Rutkai and colleagues specifically demonstrate impaired mitochondrial oxygen consumption rate after stroke. A better reference for restoring mitochondrial function in stroke may be that of Nguyen et al (Eye Opener in Stroke. Stroke. 2019;50(8):2197-2206. doi:10.1161/STROKEAHA.119.025249).

Response: We are sorry for missing the important reference. We have now added this reference in the revised Introduction section.

4. Methods and Results: Briefly stating the possible reason for why the 14% neurons escaped AM-mito coating is welcomed.

Response: Thank you for raising the important point regarding the coating efficacy. To be honest, it is really hard to explain why some mitochondria did not show evans blue signal following coating process while two reasons may be possible. One can be less amount of evans blue incorporation that weakens evans blue signal in flow cytometry analysis. The other reason can be unexpected failure to incorporate mitochondria. We now added the explanation in the revised Result section.

5. Also, it will be interesting if there's a way to delineate whether OGD preferentially damaged neurons which were not AM-mito coated as opposed to those that were AM-mito may be interesting as this will further support the claim that the AM-mito technology confers neuroprotection.

Response: Thank you for raising the important point. We agree with the reviewer that it remains unclear whether incorporation of AM-mito into neurons selectively protects neurons following ischemic injury. We now tried our best to investigate whether AM-mito-transferred neurons survived at 24 hours after 2h-OGD challenge. Intriguingly, Mitotracker Green-labeled AM-mito (10 µg per

well) were found in approximately 55% of total neurons at 24 hrs after OGD and 46% or 8.7% of AM-mito+ neurons were survived or dying/dead, respectively. This may further support that AM-mito incorporation into neurons may support neuroprotection as a consequent. We now included the data in the revised Figure 3f.

6. Discussion: The statement "It has reported that extracellular mitochondria can enter cells via endocytosis 29, integrin-dependent mechanisms 13 or macro-pinocytosis 28" may need to be re-written as follows: ". Among the many postulated mechanisms of neuronal internalization of mitochondria include extracellular mitochondria can enter cells via endocytosis 29, integrin-dependent mechanisms 13 or macro-pinocytosis 28."

Response: We apologize and thank you for the suggestion. We now rewrote the sentence as suggested in the revised Discussion.

Reviewer #2:

1. Fig. 1c, what is the criteria to judge the optimal ratio for DOTAP:DOPE? The authors did not explain how to read the data, is any kind of quantitation needed here?

Response: We apologize for being unclear on Fig 1c of our paper, and thank the reviewer for giving us another opportunity to clarify our intent. To determine optimal ratio and concentration of DOTAP:DOPE, we initially tested evans blue encapsulation. The criteria was to minimize the leak of evans blue into PBS solution. When the encapsulation was unsuccessful, evans blue visually leaked into PBS solution along with smaller pellets while optimal ratio and concentration cleared evans blue leak along with accumulation of larger pellets in the bottom of tube. We now modified Fig 1c to emphasize our criteria on determination of successful encapsulation.

2. Fig. 2d, to prove the mitochondria itself was not changed after surface modification, the study used some mitochondria protein as the readout. Is there any other functional readout, like ATP synthesize, morphology changes to be detected?

Response: Thank you for raising this point. In our revised manuscript, we measured ATP and attempted TEM imaging. We could confirm that there was no difference in ATP content between control mitochondria and coated mitochondria. Furthermore, TEM analysis also confirmed that there was no significant morphology changed in mitochondria after coating. We now added these new experiments to Figure 2.

3. Fig. 3b, it would be nice to present a quantitation here to better show the increased intracellular mitochondria. Moreover, a negative control that treatment of only coating material (DOTAP/DOPE) is needed here to show the effect of those biomaterials on intracellular endogenous mitochondria.

Response: Thank you for raising these two points.

As for the quantification, we now quantified the fluorescent intensity after mitochondria treatment in the recipient neurons and the data was included in Figure 3b together with the images.

Thank you for the suggestion about a negative control. We have tested whether DOTAP/DOPE coating itself induced neuroprotection. However, we did not find any neuroprotective effect when DOTAP/DOPE-coated liposomes were treated in neurons following 2h OGD. The data is shown in Figure 3e.

4. Fig. 3d, what is WST assay stand for? There is not any full name presented throughout the manuscript.

Response: We are sorry for being unclear. WST stands for Water-soluble tetrazolium salts. We now included the full name in the revised main text and Method section.

5. Since the ultimate goal of this study is for the intervention of the disease, it would be nice to talk about their thoughts about in vivo delivery of mitochondria into specific region and cell types in the discussion section.

Response: Thank you for raising the very important point. We agree that it is important to address how DOTAP/DOPE-coated mitochondria might be useful when it comes to the intervention of CNS injury such as ischemic stroke. In this revision, we tried our best to test the neuroprotective ability of AM-mito in a mouse model of transient focal cerebral ischemia as our previous study (Nakamura et al., Stroke. 2020 Oct;51 (10):3142-3146). AM-mitochondria (100 ug) or control-mitochondria (100 ug) were administered intravenously and we found that AM-mito improved the accumulation to the brain along with reducing infarction. We now included the data in the new Figure 3g-i.

We also added discussion as follows;

Fifth, although mitochondrial allograft may explicit beneficial protection under pathophysiological conditions, targeting specific tissues or cells are still challenging. The mitochondrial matrix is highly negatively charged ³¹, suggesting that transplanted mitochondria may be attracted to acidotic ischemic tissue ³². But what remains to be elucidated is how to target dying cells that should be selectively rescued. It has been considered that the exposure of phosphatidylserine (PS) on the outer plasma membrane are a hallmark of dying cells. Because PS is negatively charged, mitochondria coated by cationic artificial membrane might support electrostatic interaction. Further studies are warranted to rigorously validate mitochondria migration and incorporation into brain cells such as neurons, astrocytes, microglia, and endothelial cells under pathophysiological conditions in the CNS.

6. Minor point: Main text line 4, is there a typo “and progression of cell death and 5”

Response: We are sorry for the typo. We now corrected the sentence.

Reviewer #3:

1. Fig.2d should include mitochondria internal reference protein such as COX IV in order to confirm that the total amount of mitochondria is the same in both groups.

Response: Thank you for raising the point. We included COX IV WB in Fig. 2d.

2. It is better that Fig.3c add statistic diagram to compare the transfer efficiency of Cont-mito and AM-mito.

Response: Thank you for the suggestion. We included quantification data and p values in Fig. 3c.

3. Please add the specific methods of extracting and isolating mitochondria in the part of Materials and Methods.

Response: We are so sorry for missing the detail of mitochondria isolation. We now included it in the revised Method section.

4. Mitochondrial transplantation as a potential neuroprotective therapy, the safety problems associated with their mitochondrial preservation and transport in vitro cannot be ignored. Can the surface coating of artificial lipid membrane improve the stability of mitochondria in vitro? An explanation or argument can be added in the "discussion" section.

Response: Thank you for the suggestion. We agree that mitochondrial preservation is critical to increase flexibility of mitochondria transplantation therapy. In this study, DOTAP/DOPE-coated mitochondria showed higher membrane potential, better incorporation into cells in accompanied by more remaining in the ipsilateral hemisphere compared to non-coated control mitochondria, implicating that MitoCoat may stabilize isolated mitochondrial function. Regarding clinical translation, mitochondria preservation for long-term storage significantly increases flexibility of the transplantation therapy yet it is the biggest challenge. Until now, it has been reported that glycerol, DMSO, or trehalose may preserve mitochondrial integrity following freeze/thaw process and these traditional preservation methods may be utilized with MitoCoat for long-term storage until the usage. We now explicitly discussed about the issue in the revised Discussion section.

5. Higher concentration of isolated mitochondria may slightly increase neuron death. While improving the transport efficiency, may AM-mito also have potential neurotoxicity? In Fig.3d, the group with a higher concentration of AM-mito can be added to discuss this issue.

Response: Thank you for the suggestion. We now repeated a key experiment to test the highest concentration to determine neurotoxicity of AM-mito. Importantly, we found that the highest concentration 100 μ g of AM-mito per well still induced neuroprotection while the efficacy was slightly lower than 50 μ g per well. We now explicitly discussed that non-coated mitochondria may induce neurotoxicity while it is also important to clarify the optimal concentration of AM-mito to maximize beneficial protection in future studies.

6. In addition to the caveats listed in the discussion, the application mode of coated mitochondria should be explored. Different from endogenous mitochondrial transfer, it is difficult to fulfill the transplantation of exogenous liposome coated mitochondria to targeted tissue or cells. It is recommended to add this part to the discussion.

Response: Thank you for raising the very important point. We totally agree that it is important to discuss how DOTAP/DOPE-coated mitochondria might be useful when it comes to the intervention of the disease. As we noted earlier section above, we added neuroprotection study in the revised Figure 3 and also discussed the issue in the discussion section as follows;

Fifth, although mitochondrial allograft may explicit beneficial protection under pathophysiological conditions, targeting specific tissues or cells are still challenging. The mitochondrial matrix is highly negatively charged³¹, suggesting that transplanted mitochondria may be attracted to acidotic ischemic tissue³². But what remains to be elucidated is how to target dying cells that should be selectively rescued. It has been considered that the exposure of phosphatidylserine (PS) on the outer plasma membrane are a hallmark of dying cells. Because PS is negatively charged, mitochondria coated by cationic artificial membrane might support electrostatic interaction. Further studies are warranted to rigorously validate mitochondria migration and incorporation into brain cells such as neurons, astrocytes, microglia, and endothelial cells under pathophysiological conditions in the CNS.

7. The resolution of figures (the text part mainly) is low, especially in Fig.2a and Fig.2e.

Response: Thank you for the suggestion. We now modified the text part in Fig. 2a and new Fig. 2g, accordingly.

REVIEWERS' COMMENTS:

Reviewer #1 (Remarks to the Author):

The authors have fully addressed my original suggestions. I have no further comments. This study will be well received in the field of stroke neuroprotection and the general area of neurological disorders and regenerative medicine.

Reviewer #2 (Remarks to the Author):

The revised manuscript by Nakano and colleagues has been greatly improved by replacing some high resolution images, adding necessary controls and additional supporting evidence data to support that their new method is valid and has the potential for translational use. They have addressed my previous concerns, and I would recommend its publication.

Reviewer #3 (Remarks to the Author):

Overall, the authors did a very good job of addressing concerns and adding data/supporting information where we asked. I believe this work makes important contributions to the field. Now this manuscript is suitable for publication.